# Pilot Study for Immunogenicity of SARS-CoV-2 Vaccine with Seasonal Influenza and Pertussis Vaccines in Pregnant Women

**DOI:** 10.3390/vaccines11010119

**Published:** 2023-01-03

**Authors:** Ching-Ju Shen, Yen-Pin Lin, Shu-Yu Hu, Ching-Fen Shen, Hui-Yu Chuang, Chin-Ru Ker, Der-Ji Sun, Yu-Hsuan Yang, Chao-Min Cheng

**Affiliations:** 1Department of Obstetrics and Gynecology, Kaohsiung Medical University Hospital, Kaohsiung Medical University, Kaohsiung 807, Taiwan; 2Institute of Biomedical Engineering, National Tsing Hua University, Hsinchu 300, Taiwan; 3Department of Pediatrics, National Cheng Kung University Hospital, College of Medicine, National Cheng Kung University, Tainan 704, Taiwan; 4Graduate Institute of Clinical Medicine, College of Medicine, Kaohsiung Medical University, Kaohsiung 807, Taiwan; 5Department of Obstetrics and Gynecology, Pojen Hospital, Kaohsiung 804, Taiwan; 6Department of Post Baccalaureate Medicine, Kaohsiung Medical University, Kaohsiung 807, Taiwan

**Keywords:** SARS-CoV-2 vaccine, influenza, pertussis, neutralizing antibody, immunogenicity

## Abstract

Background: It is well known that the implementation of routine immunizations to prevent vaccine-preventable diseases has a significant impact on the health and well-being of infants, children, and pregnant women. We aimed to evaluate the influence of influenza, tetanus toxoid, reduced diphtheria toxoid, and acellular pertussis (Tdap) vaccine on the immunogenicity of SARS-CoV-2 vaccine among pregnant women, the priority population recommended for vaccination. Methods: We conducted a prospective study among pregnant women without previous SARS-CoV-2 infection in Taiwan. Maternal and umbilical cord blood samples at delivery were analyzed for the percentage of inhibition of neutralizing antibodies (NAbs) against the original strain, Delta, and Omicron variants of SARS-CoV-2 as well as the total antibody to the SARS-CoV-2 spike protein. We examined the association between different doses of SARS-CoV-2 vaccine in combination with influenza and Tdap vaccination, and two-dose SARS-CoV-2 vaccination with or without influenza and Tdap vaccines via a two-sample *t*-test. Results of *p* < 0.05 were considered to be statistically significant. Results: 98 pregnant women were enrolled in our study, with 32 receiving two doses of SARS-CoV-2 mRNA-1273 vaccine, 60 receiving three-dose of mRNA-1273, and 6 receiving one-dose of ChAdOx1 and two-dose of mRNA-1273. Twenty-one participants were immunized with SARS-CoV-2, influenza, and Tdap vaccines. Of these 21 individuals, there were no significant NAbs levels in maternal and cord blood samples against the Omicron variant, regardless of doses or type of SARS-CoV-2 vaccine. However, antibody responses against the wild-type and Delta variant were significantly lower in all maternal sera in the two-dose SARS-CoV-2 vaccine group. Among 32 women receiving two-dose mRNA-1273, significantly lower levels of NAbs in maternal sera were observed against the Delta variant and total antibody both in maternal sera and cord blood were observed in individuals receiving SARS-CoV-2 and influenza vaccine. Conclusion: This is the pilot study to demonstrate the effects of influenza and the Tdap vaccine on the immunogenicity of the SARS-CoV-2 vaccine among pregnant women. These results suggest that combination vaccination during pregnancy may result in immunogenic interactions.

## 1. Introduction

On 11 November 2021, the first Omicron variant SARS-CoV-2 case was reported in Botswana [1], and 2 weeks later, the WHO declared this novel variant the fifth SARS-CoV-2 variant of concern (VOC) and officially labeled it as the Omicron variant (B.1.1.529) [2]. Many spike protein mutations have been identified in the Omicron variant, associated with increased transmissibility, decreased effectiveness of therapeutic monoclonal antibodies, and immune evasion after natural infection and vaccination [3,4]. This led to increased breakthrough infections in previously infected and vaccinated individuals [5]. The first confirmed case of the Omicron variant detected in the United States was reported on 1 December 2021. More than 38,000 pregnant women were infected during the Omicron surge at the end of May 2022 [6]. It is worth noting that, regarding the development of coronavirus disease 2019 (COVID-19) vaccines, there is limited data on safety and effectiveness in pregnant women excluded from clinical trials.

Even though available evidence suggests that SARS-CoV-2 infection during pregnancy is related to a higher risk of complications and death than in non-pregnant women, the lack of relevant information makes pregnant women vaccination-reluctant. Neither the American College of Obstetricians and Gynecologists (ACOG) nor the Society for Maternal-Fetal Medicine (SMFM) actively recommended vaccination of pregnant women until 30 July [7], after the first real-world vaccine studies had been published [8]. Maternal immunization has the potential to protect mothers and young infants from specific infectious diseases. Infants born to vaccinated mothers have placenta-transferred antibodies that provide passive immunity until they develop their immunity through routine vaccinations. Doctors routinely recommend that pregnant women receive pertussis and influenza vaccinations [9]. Based on ACOG’s recommendations, COVID-19 vaccines may be administered simultaneously with other vaccines, including influenza, tetanus toxoid, reduced diphtheria toxoid, and acellular pertussis (Tdap) vaccines. All statements supported the necessity of vaccination against infectious diseases in pregnancy and the benefit for newborns in the COVID-19 pandemic. However, most COVID-19 vaccine trials excluded participants receiving other vaccinations at the time of injection with study vaccines, so interaction data has been poorly addressed, and vaccine administration has not factored in co-vaccination. Pregnant individuals are a unique subgroup of the population with distinctive considerations regarding themselves and their fetus/newborn, which leads them to administer multiple vaccines within the gestational period during the COVID-19 epidemic. But there is insufficient data regarding whether or not vaccines administered during pregnancy alter COVID-19 vaccine immunogenicity. Furthermore, choosing reliable serological tests to measure individual immunity is critical. Detecting total binding antibodies does not equal protective immunity. Unlike IgM and IgG in the isotype-specific enzyme-linked immunosorbent assay, neutralizing antibody titer is more valid in assessing humoral protection and vaccine efficacy [10]. In this study, we chose the pseudovirus-based virus neutralization test which is highly predictive of immune protection to investigate the effects of the influenza vaccine (Vaxigrip Tetra, inactivated quadrivalent influenza vaccine) and Tdap vaccine (Adacel, 5-component pertussis vaccine) on the immunogenicity of co-administered SARS-CoV-2 vaccine (either an adenovirus viral vector COVID-19 vaccine, ChAdOx1 or an RNA COVID-19 vaccine, mRNA-1273).

## 2. Materials and Methods

### 2.1. Study Design and Participants

We conducted a prospective cohort study of Taiwanese pregnant women ≥20 years of age who received at least two doses of the COVID-19 vaccine (ChAdOx1; Oxford-AstraZeneca or mRNA-1273; Moderna). The Kaohsiung Medical University Hospital review board approved this study (IRB number: KMUHIRB-SV(II)-20210087); participants provided written informed consent. In prenatal care, health caregivers provided adequate vaccine information, including the public-funded COVID-19 vaccine, influenza vaccine, and self-paid pertussis vaccine, and encouraged their vaccination. Whether to vaccinate or not depended on the individual health choice. Maternal pertussis vaccination was between 28–35 weeks of gestation. Flu shots were given in any trimester when the vaccine was available. Eligible participants were healthy without pregnancy-associated complications, had no contraindications to vaccination, and were confirmed negative for SARS-CoV-2 infection via nasopharyngeal swab reverse transcription-polymerase chain reaction within two days of admission. Individuals with intrapartum complications were excluded from the study. All participants provided peripheral blood and infant cord blood collected after clamping.

### 2.2. Statistical Analysis

We analyzed maternal and cord blood to determine whether or not there were any comparative differences in the delivery efficiency of neutralizing antibodies against different variants following the administration of varying vaccine combinations. This data was analyzed using GraphPad Prism (GraphPad Software, San Diego, CA, USA). A sample *t*-test (between the two different sets of data) was performed to compare rates of neutralizing antibody inhibition in maternal blood and cord blood from subjects receiving different vaccine combinations, such as two or three doses of COVID-19 vaccine, three doses of Modena, or two doses of AZ plus one dose of Modena, with or without Tdap or Flu vaccination. Results of *p* < 0.05 were considered to be statistically significant.

### 2.3. Neutralizing Antibody Inhibition Test of SARS-CoV-2 Omicron, Delta, and Wildtype Variants

As described in the experimental design, we selected the GenScript cPass SARS-CoV-2 Neutralization Antibody Detection Kit, Acro Anti-SARS-CoV-2 (B.1.1.529) Neutralizing Antibody Titer Serologic Assay Kit (Spike RBD), and AdipoGen Delta SARS-CoV-2 Neutralizing Antibody Detection Kit for the detection of neutralizing antibodies against SARS-CoV-2 spike RBD in samples by competitive ELISA. The 96-well plates in the kit were pre-coated with human ACE2 protein. To begin the experiment, we added serum samples, positive control, and negative control to the 96-well plate, followed by the addition of HRP-SARS-CoV-2 spike RBD. After incubation, the plate was washed, and substrate was added to the plate. Finally, a stop solution was added to stop the reaction, and the absorbance intensity was measured at 450 nm.

The presence of neutralizing antibodies in the sample competed with ACE2 for HRP-SARS-CoV-2 spike RBD binding. The intensity of the detected signal was considered proportional to the concentration of neutralizing antibodies against SARS-CoV-2. If there was no neutralizing antibody in the sample, HRP-RBD bound ACE2 and was colored using TMB. Conversely, if there was neutralizing antibody in the sample, the absorbance decreased. Positive control and negative control were attached to the set. If the absorbance value of the sample was less than 0.8 times the negative control, the sample was defined as “with neutralizing antibody”. Absorbance values at 450 nm were measured using a microplate spectrophotometer (Molecular Devices, San Jose, CA, USA). The obtained O.D value results were used to calculate the percent inhibition of neutralizing antibodies based on the following formula,
$$\mathrm{Inhibition}\;\%\;=\left(\;1-\frac{\mathrm{OD}450\;\mathrm{value}\;\mathrm{of}\;\mathrm{sample}}{\mathrm{average}\;\mathrm{OD}450\;\mathrm{value}\;\mathrm{of}\;\mathrm{negative}\;\mathrm{control}}\right)\times 100\%$$

### 2.4. Anti-SARS-CoV-2 S Protein Total Antibody

Quantitative determination of pan-immunoglobulin to RBD of SARS-CoV-2 spike protein was achieved via automated electrochemiluminescence immunoassay using a Roche Elecsys anti-SARS-CoV-2 S (Roche Diagnostics, Basel, Switzerland; hereafter called Roche S). As previously described, the assay was performed on a Roche Cobas e601 system (Roche Diagnostics) and used plasma or serum from vaccinated volunteers for measurement [11]. This assay has a measuring range of 0.40 to 250 U/mL (up to 2500 U/mL with on-board 1:10 dilution). A concentration of <0.80 U/mL was considered negative and ≥0.80 U/mL was considered positive [12].

## 3. Results

Among the 98 participants, 32 (32.65%) received 2 primary doses of COVID-19 vaccine; 66 (67.35%) received a booster shot. Patient pregnancy characteristics, newborn characteristics, and the interval between vaccination to delivery are shown in Table 1. In the two-dose COVID-19 vaccine group, all received the Moderna vaccine, and doses were given at least 4 weeks apart. Of these women, 8 received influenza and pertussis vaccination, 8 received influenza vaccine only, 8 received pertussis vaccine only, and 8 received neither influenza nor pertussis. The median interval between the second dose and delivery was 9.84 weeks (IQR: 12–8). In the three-dose COVID-19 vaccine group, 13 received both influenza and Tdap vaccination, and 53 received Tdap only. The median interval between the third dose and delivery was 5.93 weeks (IQR: 7.5–4). Of the women with COVID-19, influenza, and Tdap vaccine administration, 6 were vaccinated with the Oxford-AstraZeneca vaccine as the primary two-dose series, followed by the Moderna vaccine, and 7 received 3 Moderna vaccine immunizations. The minimum interval between the second dose and the third dose was 12 weeks. All cases tested negative for SARS-CoV-2 nucleocapsid IgG.

In subjects receiving COVID-19 plus influenza and Tdap vaccines, the median percentage of NAbs inhibition against SARS-CoV-2 Omicron variant in maternal sera was lower in two-dose AZ plus Moderna and two-dose Moderna groups compared with three-dose Moderna (7.54% versus 4.07% versus 21.07%), but differences were not statistically significant (*p* = 0.25 and 0.07) (Figure 1). The results were similar in cord blood for each group (8.6% versus 11.9% versus 25.94%), which were also not statistically significant (Figure 1). The median cord-to-maternal ratio of Omicron variant NAbs in each group was 0.94, 1.43, and 1.06, respectively. For the Dela variant, the percentage in the three-dose group was not significantly different for either combination of AZ plus Moderna or Moderna vaccines (in maternal sera, *p* = 0.16; in cord blood, *p* = 0.53) (Figure 2). However, in the two-dose group, a significantly lower percentage of NAbs inhibition was found in maternal sera and cord blood (*p* < 0.05) (Figure 2). Figure 3 shows that the percentage of NAbs inhibition percentage in maternal sera for the wild type in the two-dose group was significantly lower compared to that of the three-dose Moderna group. This difference was not found in cord blood. In addition, there was no difference in NAbs titer in the three-dose group, regardless of the vaccine type. The median interval between the third booster shot to delivery in the three-dose Moderna group was 4.71 weeks, while that for the one-dose AZ plus two-dose Moderna group was 8.33 weeks.

NAbs for the Omicron variant among the two-dose Moderna vaccine recipients were far below the cutoff value (30%) in maternal sera and cord blood with or without co-administration of influenza and Tdap vaccines (Appendix A). These results indicated that the neutralizing antibodies to SARS-CoV-2 were not detected. Being vaccinated with two-dose Moderna and influenza vaccines was associated with statistically significantly lower maternal NAbs levels against the Delta variant compared to receiving only the Moderna vaccine (Figure 4). This trend was also found in umbilical cord blood, but it was not statistically significant. For the wild-type variant, there were no statistically significant differences in percentage inhibition of NAbs level in maternal serum and cord blood in the two-dose Moderna with or without influenza vaccination (Figure 5). We also quantitatively assessed the amount of anti-SARS-CoV-2 antibodies for each sample. In the three-dose Moderna and two-dose AZ plus one-dose Moderna vaccination groups, antibody concentrations of >2500 U/mL were 85.71% and 50% in maternal blood and 71.43% and 83.33% in cord blood, respectively (Appendix A). In each of the analyzed 4 subgroups receiving the two-dose COVID-19 vaccine, there was a significantly lower titer of total antibodies against SARS-CoV-2 in individuals vaccinated with COVID-19 and influenza vaccine (Figure 6). Figure 7 presents the percentage of inhibition of NAbs in the three-dose group against different Omicron variants. Twelve weeks after the booster dose with Moderna in Tdap vaccinated individuals, as compared with the median titer against the wild type, the median titers in maternal blood were 49.96%, 28.56%, and 20.86% against the BA.1, BA.2, and BA.5 subvariants. The antibody responses were similar in cord blood, with 51.67%, 25.27%, and 18.05% against the BA.1, BA.2, and BA.5 subvariants.

## 4. Discussion

SARS-CoV-2 infection in pregnant women increased the risk for poor perinatal outcomes and maternal mortality through direct and indirect effects on the fetus and placenta [13,14]. New variants, such as Delta and Omicron, have more aggressive, transmissible, and vaccine-resistant [15]. Fortunately, while the prevalence of infection increased during the Omicron wave of SARS-CoV-2, there were fewer cases with severe complications in pregnant women compared to pre-Delta and Delta waves [16]. Improved outcomes do not imply that the Omicron variant is less threatening to pregnant women but rather reduced disease severity was strongly related to increased vaccine coverage. A study showed that the severity and outcomes were similar to those in the pre-Delta period for unvaccinated pregnant women [17]. Although vaccinated pregnant women had lower clinical severity compared with the unvaccinated group [18]. Protection against SARS-CoV-2 infection after two doses of vaccine wanes over time [19], while booster doses have been recommended for pregnant women to protect against new variants and offer neonates better transplacental antibody transfer [20,21,22]. However, data on the immune response following SARS-CoV-2 vaccine administration in pregnant women who received other vaccinations are scarce.

Our study examined the effects of influenza and pertussis vaccines on the antibody response of two different SARS-CoV-2 vaccines. The percentages of NAbs inhibition against the wild-type, Delta, and Omicron variants in maternal serum and cord blood were higher in the three-dose Moderna group compared with the two-dose group, indicating the values of a booster dose on immunogenicity. For participants who received a booster shot, three doses of Moderna appeared to drive higher NAbs levels in both maternal and cord blood than in the two-dose AZ plus one-dose Moderna group, but there was no statistically significant difference. These results were partially consistent with the previous study published in March 2022 [23]. The reason for the lower NAbs level is likely independent of the primary immunization type. However, it might be related to the interval between the administration of the third dose and delivery (average interval of 4.71 weeks in the three-dose Moderna group, 8.33 weeks in the two-dose AZ plus one-dose Moderna group), and the NAbs level seemed to wane rapidly after booster vaccination. Our data also indicate a reduced neutralizing antibody response to the Omicron variant compared to the response against the wild-type strain of SARS-CoV-2 or the Delta variant among vaccinated pregnant women, although booster doses improved neutralizing activity. The vaccination-mediated antibody response for the Omicron sublineage, BA.5, was weaker than BA.1 and BA.2, raising the possibility of transmission. Therefore, it is urgent to upgrade the COVID-19 vaccine and expand it for use in eligible groups to suppress the resurgence of the epidemic. On 31 August 2022, the U.S. Food and Drug Administration amended the emergency use authorizations (EUAs) of bivalent Pfizer and Moderna booster shots that target both the original COVID-19 strain as well as the BA.4 and BA.5 Omicron subvariants. This updated version of the bivalent COVID-19 vaccine is expected to provide increased protection against the currently circulating omicron variant. However, there is no real-world data on the effectiveness of these new vaccines for pregnant women.

Lazarus et al. analyzed the immunogenicity of concomitant administration of the SARS-CoV-2 vaccine (ChAdOx1 or BNT162b2) and seasonal influenza vaccine and found that non-inferiority was indicated in concomitant vaccination cohorts [24]. Toback et al. studied the immunogenicity of COVID-19 vaccine (NVX-CoV2373) co-administered with cellular quadrivalent seasonal influenza vaccines and revealed a significant geometric mean ELISA unit difference to NVX-CoV2373 between the group receiving concomitant vaccination versus the group receiving SARS-CoV-2 vaccine alone, with a geometric mean ratio of 0.57 (95% CI 0.47–0.70) [25]. Our study subjects were quite different from those of Lazarus et al. because modifications occur in virtually every facet of the immune response during a normal pregnancy. Changes in circulating immune cells, leukocytes, can lead to innate and adaptive immune dysregulation and may have an effect on the immunological response after vaccination. In our study, participants received SARS-CoV-2, pertussis, and influenza vaccination with at least one week of separation rather than coadministration. We observed the impact of influenza vaccine administration on the percentage of NAbs inhibition for the Delta variant variants in maternal sera (37.02% reduction compared with the two-dose COVID-19 vaccine only). For total antibody titer, it was also found that the concentration was significantly lower in this group than in other groups, whether in maternal or umbilical cord blood. This reduction was not apparent when assessing the immunogenicity of COVID-19 with pertussis vaccination, which is consistent with our previous findings [26]. Interestingly, the neutralizing response was not decreased with the administration of three studied vaccines during pregnancy in this cohort. It is unclear whether this difference was due to vaccine interference or the non-randomized nature of the studied groups and why this would only be true for the influenza vaccine and not the pertussis vaccine. In our study, it remains a considerable challenge to interpret the correlation between the reduction of neutralizing response with vaccine effectiveness. The impact of humoral response to the SARS-CoV-2 vaccine in combination with other vaccines in pregnancy may be related to the following hypotheses: (1) carrier-induced epitopic suppression [27,28]; (2) the effects on antigen capture, processing, or presentation [27]; or, (3) interference with lymphocyte recognition and responses [27]. Moderna mRNA-1273, a messenger RNA vaccine, uses ionizable lipid nanoparticles to deliver nucleoside-modified mRNA encoding the full-length spike protein of SARS-CoV-2 [29], and its adjuvanticity seems more complex than traditional vaccines. Although our research points out that participants who received the COVID-19 and influenza vaccines had a lower percentage of inhibition of neutralizing antibodies for the wild-type, Delta, and Omicron variants, only the level in maternal blood for the Delta variant showed statistical significance. Neutralizing antibodies against the Omicron variant were below the threshold, making it impossible to compare the differences between co-administered vaccines. Therefore, our findings should not be misinterpreted as discouraging pregnant women from getting the COVID-19 and influenza vaccines during pregnancy. More research is needed to explore the mechanism to realize the interaction between mRNA and non-mRNA vaccines.

We acknowledge some limitations of this study. The first limitation is the small number of specific-group participants. With efforts of the Advisory Committee on Immunization Practices (ACIP) and obstetric health workers, influenza and Tdap vaccination were well accepted by pregnant women. Knowledge of the availability of vaccination during pregnancy is a prerequisite for women to accept vaccination or not. All participants in this study were advised to receive routine maternal vaccination by their healthcare providers. Therefore, the number of women who receive only the influenza vaccine or Tdap vaccine will be limited. Most cases of refusal to administer the flu shot or pertussis vaccine were because of vaccine fatigue, not the side effects of vaccines, and believed their adherence to personal protective measures was sufficient to prevent infection. The second limitation is that the timing of influenza vaccine administration is not taken into account. In this study, women received influenza vaccine administration in their second and third trimesters, but we did not further analyze the sequence and interval of influenza, pertussis, and COVID-19 vaccination. A third limitation of this study is that we did not further explore cellular immunity. Cellular immune responses are likely to play an important role in disease severity. Further information on immunity research will be summarized in our next study. Lastly, this study was an open-label, non-randomized trial design. Implementation of maternal immunization is supported by global health policy; therefore, an important ethical issue arises if this study is designed as a randomized controlled trial. The strength of this study is that we are few focused on the immune response of maternal routine vaccination during the epidemic. We preliminarily indicated the impact of influenza and Tdap vaccination on the production of antibodies after COVID-19 vaccine administration in pregnant women, but the data in this field were still scarce. Depending on the results of this pilot study, further large-scale main studies are regarded to be feasible and valuable. We would further investigate the mechanism of immune interference of the COVID-19 and influenza vaccines in pregnancy. Second, we would demonstrate the protective immunity with the inhibition rate of neutralizing antibody which is superior in sensitivity.

## 5. Conclusions

During the SARS-CoV-2 pandemic, vaccination during pregnancy is essential, as it protects adults, fetuses, and young infants. Although the percentage of neutralizing antibody inhibition against the Omicron variant significantly decreased, cellular immunity may play a critical role in protection from severe illness as antibodies wane over time. The third booster dose produces higher maternal antibodies that pass through the placenta to the fetus and remains strongly recommended during pregnancy. Our experimental results revealed that vaccination with the COVID-19 and influenza vaccine had a negative impact on the inhibition rate of neutralizing antibodies, and co-administration of pertussis vaccine seemed to reverse this effect. Although the related mechanism remains unknown, we identify this key issue needed for further investigation. There may be an interaction among vaccines administered during pregnancy, but this should not contraindicate or hinder SARS-CoV-2 vaccination. Booster jab appears to improve the neutralization titers but seems to be insufficient for new mutants of SARS-CoV-2. We strongly recommend that pregnant women should get a bivalent booster to protect them from infection. More importantly, individuals infected with SARS-CoV-2, influenza, or pertussis experienced similar clinical manifestations, making diagnosing such co-infections more challenging. Prior studies found that SARS-CoV-2 and influenza co-infection was associated with more severe clinical manifestations and increased mortality [30,31,32,33]. At the time of writing of this manuscript, Australia experienced has just experienced its worst flu season in five years, and the surge may be related to insufficient vaccine-acquired immunity. Vaccinations against SARS-CoV-2, influenza, and pertussis for all eligible individuals should continue to be encouraged to prevent single and dual infections.

## Figures and Tables

**Figure 1 vaccines-11-00119-f001:**
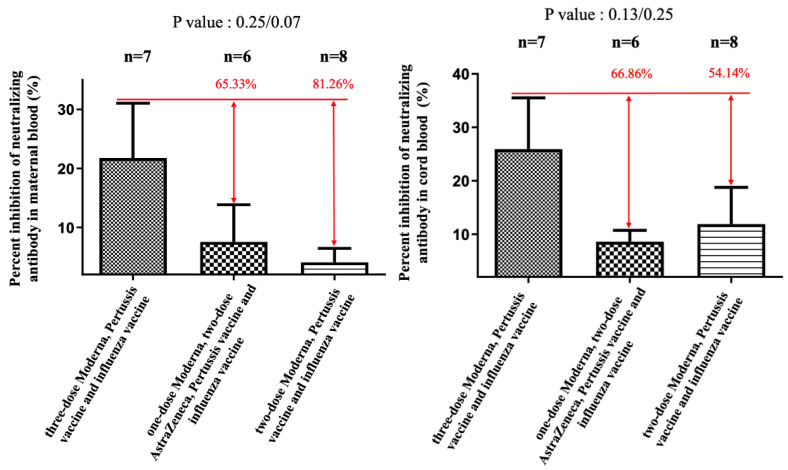
The percentage of neutralizing antibody inhibition against the Omicron variant in maternal and cord blood. In the two-dose AZ plus two-dose Moderna group, there was a 65.33% reduction (*p* = 0.25) in maternal blood and a 66.86% (*p* = 0.13) reduction in cord blood compared with the three-dose Moderna group. In the two-dose Moderna group, there was an 81.26% reduction (*p* = 0.07) in maternal blood and a 54.14% (*p* = 0.25) reduction in cord blood compared with the three-dose Moderna group.

**Figure 2 vaccines-11-00119-f002:**
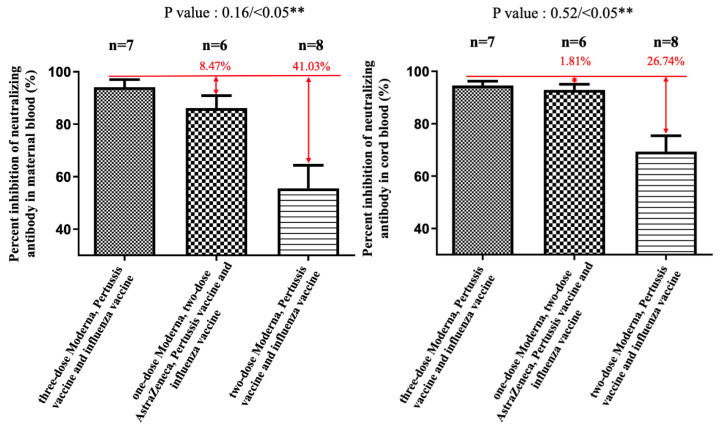
The percentage neutralizing antibody inhibition against the Delta variant in maternal and cord blood. In the two-dose AZ plus one-dose Moderna group, there was an 8.47% reduction (*p* = 0.16) in maternal blood and a 1.81% (*p* = 0.52) reduction in cord blood compared with the three-dose Moderna group. In the two-dose Moderna group, there was a 41.03% reduction (*p* < 0.05) in maternal blood and a 26.74% (*p* < 0.05) reduction in cord blood compared with the three-dose Moderna group. ** indicates statistically significant.

**Figure 3 vaccines-11-00119-f003:**
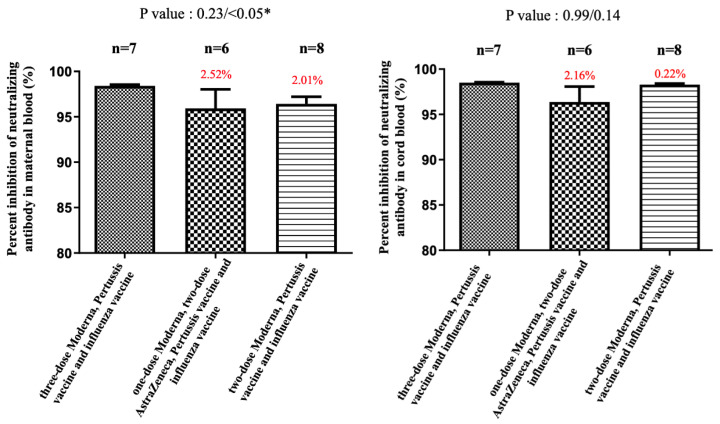
The percentage of inhibition of neutralizing antibody (%) against the wild type in maternal and cord blood. In the two AZ plus one Moderna group, there was a 2.52% reduction (*p* = 0.23) in maternal blood and a 2.16% (*p* = 0.99) reduction in cord blood compared with the three-dose Moderna. In the two-dose Moderna group, there was a 2.01% reduction (*p* < 0.05) in maternal blood and a 0.22% (*p* = 0.14) reduction in cord blood compared with the three-dose Moderna. * indicates statistically significant.

**Figure 4 vaccines-11-00119-f004:**
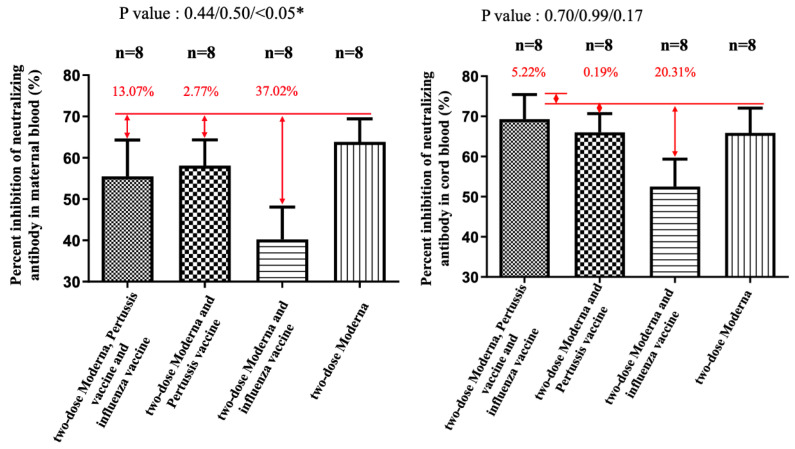
The percentage of inhibition of neutralizing antibody (%) against the Delta variant in maternal and cord blood. In the Moderna + Tdap + influenza group, there was a 13.07% reduction (*p* = 0.44) in maternal blood and a 5.22% (*p* = 0.70) increase in cord blood compared with the Moderna only. In the Moderna + Tdap group, there was a 2.77% decrease (*p* = 0.50) in maternal blood and a 0.19% (*p* = 0.99) reduction in cord blood compared with the Moderna only. In the Moderna + influenza group, there was a 37.02% decrease (*p* < 0.05) in maternal blood and a 20.31% (*p* = 0.17) reduction in cord blood compared with the Moderna only. * indicates statistically significant.

**Figure 5 vaccines-11-00119-f005:**
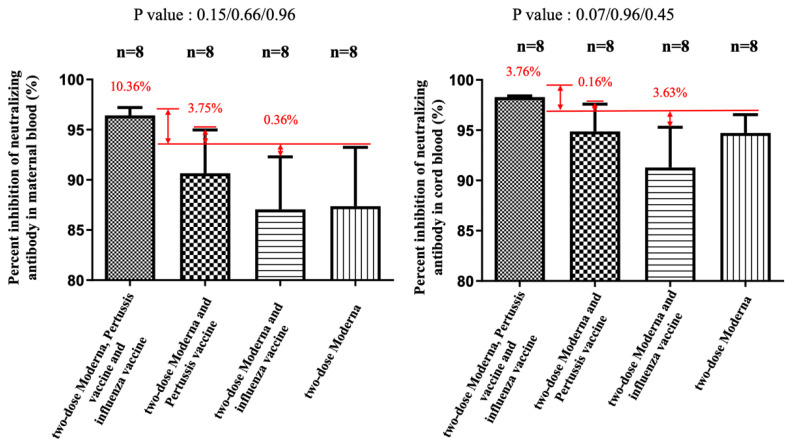
The percentage of inhibition of neutralizing antibody (%) against the wild type in maternal and cord blood. In the Moderna + Tdap + influenza group, there was a 10.36% increase (*p* = 0.15) in maternal blood and a 3.76% (*p* = 0.07) increase in cord blood compared with the Moderna only. In the Moderna + Tdap group, there was a 3.75% increase (*p* = 0.66) in maternal blood and a 0.16% (*p* = 0.96) increase in cord blood compared with the Moderna only. In the Moderna + influenza group, there was a 0.36% decrease (*p* = 0.96) in maternal blood and a 3.63% reduction (*p* = 0.45) in cord blood compared with the Moderna only.

**Figure 6 vaccines-11-00119-f006:**
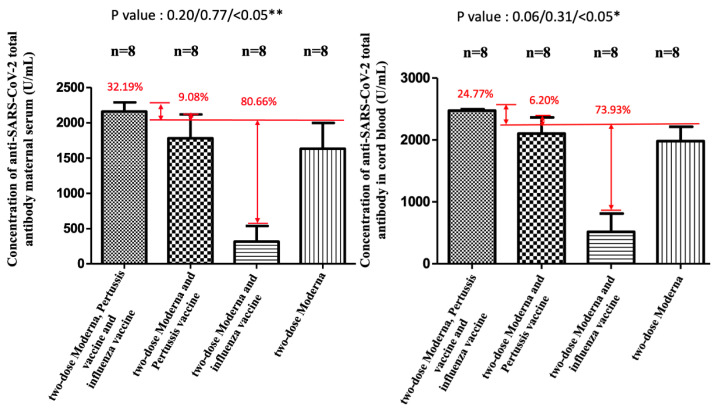
Quantitative measurement of anti-SARS-CoV-2 antibody in maternal and cord blood. ** and * both indicate statistically significant.

**Figure 7 vaccines-11-00119-f007:**
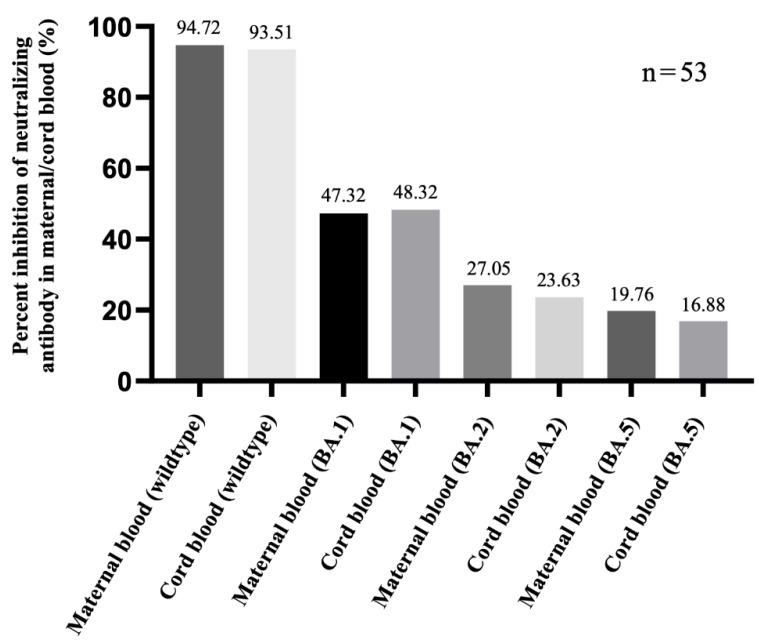
The inhibition of neutralizing antibody (%) against the Omicron sublineage in three-dose mRNA-1273 groups. The samples were tested 12 weeks after the last vaccination.

**Table 1 vaccines-11-00119-t001:** Maternal and newborn demographic and clinical data.

Variable	Included in the Analysis (Two-Dose) ^a^	Included in the Analysis (Three-Dose) ^b^
Age of mothers (years)	31.72 * (±5.27 **) IQR 34–29	33.63 * (±4.60 **) IQR 36.5–31
Parity	0.81 * IQR 2–0	1.40 * IQR 2–1
≥1		
BMI	26.76 * (±4.15 **) IQR 29.03–23.38	26.89 * (±4.62 **) IQR 30.04–24.53
Weeks of gestation at the first dose of COVID-19 vaccination (weeks)	23.5 * (±2.79 **) IQR 25.25–21	1.28 * (±9.07 **) IQR 7.5–0
Weeks of gestation at the second dose of COVID-19 vaccination (weeks)	28.75 * (±3.30 **) IQR 31–26	13.72 * (±7.38 **) IQR 19.5–7.5
Weeks of gestation at the third dose of COVID-19 vaccination (weeks)	-	32.54 * (±3.25 **) IQR 35–30.5
Interval between the third dose of COVID-19 vaccination and the collection of blood samples (day of delivery) (weeks)	-	5.93 * (±2.98 **) IQR 7.5–4
Interval between the second dose of COVID-19 vaccination and the collection of blood samples (day of delivery) (weeks)	9.84 * (±3.29 **) IQR 12–8	24.55 * (±7.37 **) IQR 31–19
Interval between the first dose of COVID-19 vaccination and the collection of blood samples (day of delivery) (weeks)	15.09 * (±2.97 **) IQR 16.5–13	36.96 * (±9.02 **) IQR 42–31
Weeks of gestation at delivery (weeks)	38.59* (±1.10 **) IQR 39–38	38.46 * (±1.17 **) IQR 39–38
Sex of newborn Male Female	15 (46.9% ***) 17 (53.1% ***)	33 (50% ***) 33 (50% ***)
Weight of newborn (g)	3039.38 * (±355.78 **) IQR 3170–2823.75	3081.42 * (±331.20 **) IQR 3310–2880

BMI: body mass index; * mean; ** standard deviation (±SD); *** percentage of all surveyed patients; ^a^ case number = 32; ^b^ case number = 66.

## Data Availability

The data of this study are available on request from the corresponding author. The data are not publicly available due to privacy or ethical restrictions.

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
