# Peer review of "Pilot Study for Immunogenicity of SARS-CoV-2 Vaccine with Seasonal Influenza and Pertussis Vaccines in Pregnant Women"

_vaccines, 2023, doi:10.3390/vaccines11010119_

Round 1

Reviewer 1 Report

This is a very important topic as vaccination during pregnancy is crucial, yet understudied. The introduction is appropriate. 

Line 93-94: instead of "depends", did the authors mean "depended" or "decided by" ?

Please elaborate on the statistical analysis. 

How was the sample size determined?

Was there a statistical power calculation performed?

Line 373-4: please rephrase the sentence. It does not seem to make sense.

The discussion is a bit long, and the first part could be shortened. The limitations are well addressed.

Reviewer 2 Report

Dear authors,

I have now completed the review of the manuscript titled "Pilot study for immunogenicity of SARS-CoV-2 vaccine with 2 seasonal influenza and pertussis vaccines in pregnant women."

In the present study, the authors evaluated the influence of influenza, tetanus toxoid, reduced diphtheria toxoid, and acellular pertussis (Tdap) vaccine on the immunogenicity of SARS-19 CoV-2 vaccine among pregnant women, the priority population recommended for vaccination. 

The manuscript is interesting and, in general, fair written.

I have some suggestions to further improve the quality of the manuscript.

1. The first section introduced some relevant articles. Please explain the results or summarize with effect sizes. 

2. The second section explains materials, especially participants. However, I was unable to find data availability statements in the article. I suggest authors clarify how other researchers can obtain the original data.

3. Authors stated that all the participants were 'healthy.' Does it mean there were no comorbidities at all? For example, hypertension, diabetes, allergy, hyperlipidemia, etc.

4. What is the future scope of the proposed research, authors have described the limitations in a good way, and I suggest that these can be the future scope of the work.
